# PhishinWebView: Analysis of Anti-Phishing Entities in Mobile Apps with WebView Targeted Phishing

## ABSTRACT

Despite the relentless efforts on developing anti-phishing techniques, phishing attacks continue to proliferate, often incorporating evasion techniques to bypass detection. While recent studies have continuously enhanced our understanding of their evasion techniques in desktop environments, few studies have been conducted to explore how the phishing attack is being handled in mobile environments, specifically WebView.

In this study, we systematically evaluate the blocking processes of anti-phishing entities in individual apps in the real world by designing the phishing attack tailored to WebView. Specifically, we select eight well-known apps using WebView, and report 80 typical phishing sites (without evasion techniques) and 130 user-agent-specific phishing sites (accessible exclusively via each app's WebView). For scalable analysis, we develop an autonomous evaluation framework and investigate accessibility of both apps and Safe Browsing entities. As a result, we find that user-agent-specific (UA-specific) phishing sites successfully evade blocking across all of the eight Android apps. We investigate accessing strategies of anti-phishing agents when trying to access UA-specific phishing sites; and only two apps find their accessible user-agents for the phishing site without any subsequent actions such as blocking the link. According to our experiment results, we present security recommendations that apps should provide users with sufficient visual clues on link previews (such as HTTP indicators and URL check status) when accessing websites through WebView. To the best of our knowledge, this is the first study that explores how the WebView environments handle phishing attacks and disclose their vulnerability in the real world.

## ACM Reference Format:
Anonymous Author(s). 2018. PhishinWebView: Analysis of Anti-Phishing Entities in Mobile Apps with WebView Targeted Phishing. In *Proceedings of Make sure to enter the correct conference title from your rights confirmation emai (Conference acronym 'XX)*. ACM, New York, NY, USA, 9 pages. https://doi.org/XXXXXXX.XXXXXXX

## 1 INTRODUCTION

Phishing attacks have been steadily increasing. According to the APWG report, from 2019 to 2022, phishing attacks has grown by more than 150% per year [1]. In 2022 alone, APWG observed a total of 1,350,037 attacks. This increase is not limited to desktop users; it

*Conference acronym 'XX, June 03–05, 2018, Woodstock, NY*
© 2018 Association for Computing Machinery.
ACM ISBN 978-1-4503-XXXX-X/18/06...$15.00
https://doi.org/XXXXXXX.XXXXXXX

also extends to mobile users. According to [16], from 2018 to 2021, South Korea recorded 38,112 incidents of messenger phishing, with losses amounting to 139.7 million US dollars. A significant portion of these incidents occurred via the KakaoTalk app [13], with 90% of all messenger phishing reported in 2019. Phishing attacks through messenger apps continue to rise year by year.

In today's real-world scenario, major web browsers on desktop and mobile platforms employ blocklisting techniques to combat phishing. However, as phishing techniques continue to evolve, attackers are employing evasion techniques to circumvent the monitoring systems of anti-phishing entities [40, 41, 48]. While desktop platforms have significantly addressed these evasion techniques, the same cannot be said for mobile devices. For example, many mobile apps are increasingly incorporating their browsers within the application itself to mitigate the inconvenience of apps having to connect to external browsers. These in-app browsers are called WebView, and starting from Android 8.0 (API level 26), Google Safe Browsing is integrated as the default feature in WebView to block malicious websites [27]. While numerous empirical analysis studies have been conducted on the blocklisting responses of various anti-phishing organizations in significant desktop and mobile browsers [32, 39, 40, 43], no prior research has been conducted on blocklisting within mobile WebView.

In this research, we comprehensively assess the real-world blocking mechanisms of anti-phishing entities within WebView applications. For this purpose, we design a user-agent-specific (UA-specific) phishing attack, which uses evasion technique and allows access only by apps' WebView user-agents. We then determine to report UA-specific phishing websites and typical phishing websites to both the app and Google Safe Browsing (GSB) to see how each responds. To select tested apps, considering the top 10 messaging and social apps by country, we select six apps (Facebook, Facebook Messenger, Instagram, Line, WeChat, and Zalo) and also the two most popular messaging and portal apps (KakaoTalk and Naver) in Korea are added in the list. Then, we develop an automated evaluator of WebView to monitor the reactions of each app at 30-minute intervals systematically. We conduct tests with a maximum of 40 domains per app at 30-minute intervals, considering the factors such as delays and interrupting behaviors of apps to restrict automation, for a 10-hour term, which is a reasonable duration considering the average age of phishing sites [42] is about 9 hours long before blocked.

As a result of our research, we discover that all UA-specific phishing websites tested on eight apps are not blocked by either app and GSB, despite reporting. After reporting, the crawler makes a total of 945 access attempts across 210 phishing websites. The crawlers attempt to access UA-specific websites with 4.23 times more IP addresses and 2.26 times more diverse user agents than when accessing typical phishing sites. Despite crawlers' numerous attempts to access, we find that out of the eight apps, only two apps access UA-specific websites using their WebView user-agent; they

also do not display warning signs or take any proactive measures. Furthermore, we find no sanctions in place when re-uploading the typical phishing links and UA-specific phishing links to all eight apps after reporting, and alerts are found on previews of phishing links. Lastly, we suggest security recommendations to provide more informative link previews showing the detailed status of linked sites. Our research findings indicate that, despite users expecting protection through applications, they may still be exposed to risks.

In summary, our contributions are as follows:

- To the best of our knowledge, our study represents the first comprehensive evaluation and analysis of anti-phishing measures in WebView environments.
- We overcome the challenges of implementing automated evaluation tools in the mobile WebView environment, conducting automated anti-phishing assessments across eight apps using Appium [2] and UiAutomator2 [3].
- Through the design of the customized phishing attack in the WebView environment, we find that anti-phishing agents in the eight apps failed to detect the attack. Additionally, our log analysis reveals the absence of anti-phishing bots utilizing WebView as a user-agent in seven of the tested apps. Even when such bots are present, they do not effectively block phishing attempts.
- We recommend implementing bots for link previews in mobile apps to provide users with advanced information about potential phishing sites, enhancing their ability to make informed decisions.

## 2 BACKGROUND

Phishing is a type of social engineering attack [35]. Phishers create fraudulent websites that imitate legitimate ones to deceive victims into providing personal information, such as account credentials or financial details, ultimately for financial gain. Nowadays, tools like phishing kits make it easy for individuals with limited knowledge of the web to create and distribute phishing sites [41]. The ease of implementing phishing attacks has led to a growing trend in phishing attacks.

### 2.1 Anti-Phishing Blocklist

Browser blocklisting [39, 40] is one of the most commonly used defense mechanisms against phishing and malicious websites in real-life scenarios. When a site is detected as a phishing site and gets blocklisted, it triggers a warning pop-up when someone tries to access it. Blocklisting applies to desktop and mobile devices, making it a versatile defense method. Two prominent anti-phishing organizations that are extensively used for blocklisting are Google Safe Browsing (GSB) [10] and Microsoft Smart Screen [17]. Most web browsers rely on these services for protection.

### 2.2 Evasion Techniques

Evasion techniques are vulnerabilities in blocklisting methods [39, 40, 49]. Phishers employ various evasion techniques to circumvent security systems like bot crawlers to remain exposed on the web for longer. These techniques can be broadly categorized into server-side and client-side approaches. Server-side evasion techniques rely on information in HTTP requests to identify users [40]. In contrast, client-side evasion is achieved through code running in the visitor's browser, often using JavaScript, to apply filters based on attributes like pop-up or mouse movement [48]. These evasion techniques allow phishing sites to linger longer online and avoid detection.

**User-Agent evasion technique.** User-Agent is a part of the HTTP request header that provides information about the operating system, device, and browser being used by the visitor (e.g., Mozilla/5.0 (Windows NT 10.0; Win64; x64) AppleWebKit/537.36 (KHTML, like Gecko) Chrome/116.0.0.0 Safari/537.36) [37] [48]. Websites that utilize user-agent evasion techniques show benign or error pages when the visitor's User-Agent includes terms commonly associated with crawlers or bots. This strategy enables them to evade access and avoid detection. However, our approach takes a different stance. Instead of blocking specific user agents, access is granted exclusively to users with the desired User-Agent conditions.

### 2.3 WebView in Apps

When users access URL links while chatting or using social networking apps, the in-app browser enables them to remain within the app interface instead of being sent to an external web browser [50]. The in-app browser enables seamless access to web content, enhancing the user experience and providing a means to integrate various web features within the app. This convenience benefits app developers by seamlessly incorporating web functionality and allowing users to access richer content and services without leaving the app. Due to these reasons, the usage of in-app browsers is on the rise, and most popular apps are incorporating in-app browsers.

**WebView.** In-app browser use WebView to display web content [44]. This class is the foundation for developers to create web browsers or incorporate online content within their apps. It is important to note that a WebView does not include the complete feature set of a fully-fledged web browser. Its primary function is just displaying web pages [47]. WebView offers various methods and functionalities, including navigating forward and backward through a browsing history, zooming in and out, performing text searches, injecting custom JavaScript code, and more. In the context of Android, different WebView components are available, including the standard WebView, Chrome Custom Tabs, and Trusted Web Activity. On the iOS platform, WebView options include UIWebView, WKWebView, and SFSafariViewController [44]. This discussion will focus on the most widely used Android WebView.

**WebView Safe Browsing.** Starting from April 2018 with WebView 66, Google Play Protect introduced Safe Browsing as the default feature for WebView. Developers of Android apps using WebView no longer need to make any adjustments to benefit from this safeguard. Safe Browsing in WebView has been available since Android 8.0 (API level 26) and utilizes the same underlying technology as Chrome on Android. When Safe Browsing is triggered, the app will display a warning and receive a network error. For apps designed for API level 27 and above, new APIs are available for customizing this behavior, allowing developers to tailor the Safe Browsing experience [27].

## 3 EXPERIMENTAL METHODOLOGY

Our goal is to measure how effectively anti-phishing entities perform detection in mobile WebView. We aim to measure reactions in

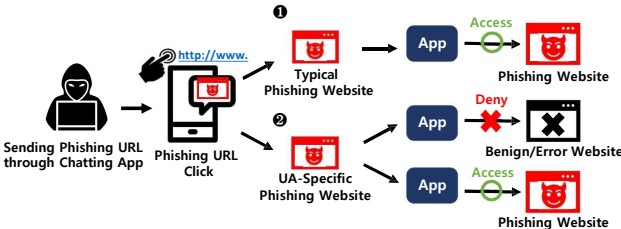

Figure 1: Tested Scenarios of Phishing Attack in WebView

| App | Service Type | Link Report | Object Reported | GSB Usage |
|---|---|---|---|---|
| Facebook | SNS | ● | Comment | ● |
| Instagram | SNS | ● | Message | ● |
| Kakaotalk | Messenger | ● | Message | ● |
| FM* | Messenger | ● | Message | ● |
| Zalo | Messenger | ○ | - | ● |
| LINE | Messenger | ● | Message | ● |
| WeChat | Messenger | ● | Message | ○ |
| Naver | Portal | ●** | Email | ○ |

*: Facebook Messenger, **: Whale Safe Browsing

**Table 1: Link Report Capability and GSB Usage of Each App**

two scenarios. First, when typical phishing sites are accessed, and second, when phishing sites specifically target apps' WebView. In the case of WebView targeting phishing sites, they restrict access to only specific apps based on the user agent in the HTTP request header. If it is not the specific app, they redirect to a benign site or display an error page, making it inaccessible. Previous studies have only evaluated blocklisting performance in mobile browsers and did not consider the apps' WebView environment [39]. Additionally, there have been challenges in using automation tools in the mobile environment, but we have addressed some of these issues.

## 3.1 Overview

*3.1.1 Attack Scenario.* We divide our scenarios into two types, as outlined in Figure 1, spreading typical phishing websites through the app and distributing WebView targeting phishing websites.
**Test Types.** In Scenario 1, phishers configure phishing sites that closely resemble legitimate websites. Then, they send links on the app's chat window or post them on social media. When the victim clicks on the URL links shared by the phishers, they are directed to the fake sites, which are difficult to distinguish from the real ones. They are prompted to enter personal information, banking details, and more, which is then sent to the phisher for exploitation. Phishing attacks within apps are indeed prevalent. For example, a victim mentioned in the [7] article was deceived by a phisher who impersonated her friend on Instagram, resulting in the theft of her Twitter and Instagram accounts.

Scenario 2 shares similarities with Scenario 1 but differs in that it involves targeting specific victims. It utilizes the user agent evasion technique described in Section 2.2, specifically targeting individuals. Apps that use WebView include unique features in their user agents, such as the app's name. Phishers leverage this feature by creating phishing sites that only grant access to users with the corresponding app's unique user agent. We call these phishing sites user-agent-specific (UA-specific) phishing sites. As in Scenario 1, phishing attackers share links through chatting or posting, attempting to lure victims. When a phisher targets an app's users who are not their intended victims, they redirect them to a benign site or display an error page, effectively blocking access. This strategy allows the phisher to persist in their attacks on their desired targets while making it difficult for anti-phishing entities to circumvent and access the site, thus enabling them to endure longer.

*3.1.2 Targeted Apps.* In 2023, among the most popular messaging apps worldwide [19], we select apps that use WebView in-app browsers. Out of the eight apps listed, five of them, namely Facebook Messenger [15], WeChat [28], LINE [14], KakaoTalk [13], and

Zalo [31], are found to use WebView. Additionally, when considering the top 10 social apps by country, Facebook [8] and Instagram [12] are consistently found to be widely used apps that utilize WebView. Therefore, we select these two as our social apps. Furthermore, we include Naver [20], the most widely used portal app in South Korea [25] as a targeted app to verify its safe browsing effectiveness. Therefore, the total number of apps to 8 are selected for our experimental purposes.
**Naver App.** According to the 2022 statistics, the Naver app ranked third among Koreans' most widely used apps, with a monthly average user base of 40 million [25]. Upon further investigation, we find that Naver utilizes its detection system called 'Whale Safe Browsing' (WSB) [29]. According to the Whale Safe Browsing announcement, it shares its database with other safe browsing systems like GSB and Phishtank [30].
**WeChat.** WeChat is the most popular messaging app in China, and it has a substantial user base in China and globally. Many apps use Google Safe Browsing to block malicious websites, but WeChat in China does not. In China, Google is blocked by their government, so they can not use Google Safe Browsing. Instead, Apple integrates the Tencent Safe Browsing service for people using Chinese IPs [26]. However, for people not using Chinese IPs, Apple uses Google Safe Browsing by default [34]. There is no known information about how it operates on Android, so we conduct experiments to uncover this.

*3.1.3 Reporting.* We mainly test two types of anti-phishing entities in the WebView environment.
**Safe Browsing Entity.** We confirm the functionality of Google Safe Browsing (GSB) within the web views of each Android app. The preliminary experiment in Section 3.2 demonstrates that, except Naver and WeChat, GSB operates effectively in the background, exhibiting the warning page consistent with those observed in the mobile Chrome browser. However, in the case of Naver and WeChat, GSB's warning pages have not been detected.
**Built-in Report.** We select messaging and social media apps that allow reporting URLs to investigate whether they have a built-in reporting feature for potentially malicious websites. As shown in Table 1, out of the seven selected apps, all except for Zalo allow users to report messages containing URLs.

## 3.2 Preliminary Test
We conduct a preliminary test to observe what apps are showing on the screen when accessing the reported phishing sites and to decide the observation period for testing attack scenarios in Section 3.1.1. We download a simple Office 365 login phishing kit sample from

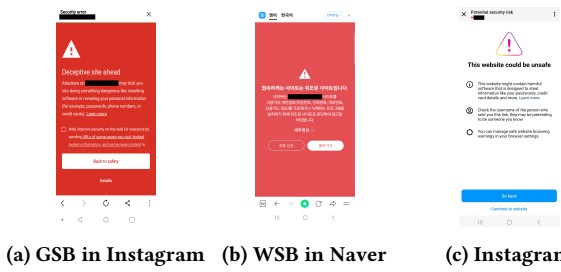

(a) GSB in Instagram  (b) WSB in Naver  (c) Instagram

**Figure 2: Warning Pages of WebView**

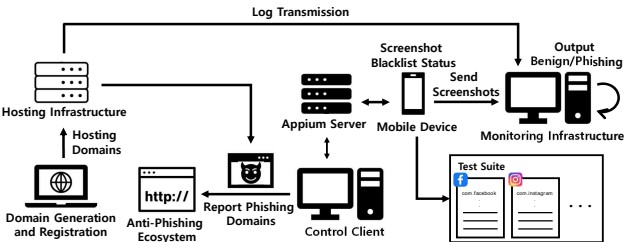

**Figure 3: Overview of Automation Framework**

phishunt.io [24] to create our test phishing sites. To avoid the inadvertent transmission of submitted information, we manually remove the backend processing of the phishing kit. Subsequently, we host a clouded Apache server containing ten random domains similar to the Office365 login page. Eight domains are configured with user-agent-specific evasion techniques, enabling access only through each app. In contrast, the remaining two domains have no evasion techniques applied, allowing all accesses. Then, we report 10 test phishing sites to GSB.

We conduct manual testing on 3 test phishing sites for each app by presenting one UA-specific phishing site and two typical phishing sites accessible through all apps every hour. In addition, to gain insights into how apps respond across different mobile platforms, we conduct tests on the latest versions, Android 12 on the Galaxy S20 and iOS 17 on the iPhone 13. We experiment, setting a time frame of one week, during which we have reported eight UA-specific phishing sites and two typical phishing sites to Google Safe Browsing. Following the research [39],we consider any response taking up to 72 hours to result in blocklisting as undetected.

**Result.** The eight UA-specific sites have not elicited GSB block response in any of the eight apps even after a week has passed. For the other two typical phishing sites, it has been evident that the GSB does not respond even after a week for the WeChat and Naver apps in the Android environment. Therefore, we have decided to report domains separately to Whale Safe Browsing and monitor Naver's response. Whale Safe Browsing collects the latest phishing data based on Naver's spam mail system and takes action by collecting phishing URLs from user reports [30]. However, even after 72 hours, domains reported to Naver have not been blocked during our experimental period. As for another app, WeChat, there is yet to be known information about the anti-phishing environment it employs outside of China for users in external countries. On the other hand, the remaining six apps, excluding these two, are all blocked within 4 hours. In the iOS environment, the GSB response is consistent with Android, with all eight apps blocking two typical phishing sites within 4 hours.

**Limitation and Additional Tests.** We have identified two key limitations in our preliminary test. Firstly, we realize that we need to incorporate the apps into the scope of safe browsing entities. During the preliminary tests, we have assumed that apps rely solely on third-party entities like Google Safe Browsing (GSB) to display phishing warning pages. However, additional testing involving access to ten online phishing sites listed on the open phishing database service, PhishTank [23], through each app reveals that each

app provides its warning page, as shown in Figure 2c, indicating that apps also have their own phishing prevention mechanisms in place. Consequently, we investigate the configuration of the built-in reporting functionalities in each app; only Zalo lacks a direct link reporting feature, providing account-level reporting as the minimum reporting unit. To further confirm this, we send a link to one test phishing site in each app and report the link by each built-in reporting functionality. After monitoring the access logs for the test phishing site for a full day, we have found no evidence of Zalo's crawler accessing the site, while other crawler bots of seven apps access our test phishing sites. In addition, the bots of three apps (Facebook, Instagram, and Facebook Messenger), which have accessed our test sites after reports, have the same IPs and user-agents. It implies that the three apps have the same phishing prevention mechanisms controlled by one entity. As a result, we have decided to add the scenario of reporting exclusively through the app in our full-scale tests.

Secondly, manually accessing the test sites through all eight apps is impractical for evaluating a large number of phishing sites. Especially given the constraints of conducting hourly access, it becomes unmanageable as the number of test domains increases. Therefore, we have decided to automate the process of accessing test phishing site links through each app in our subsequent full-scale tests.

### 3.3 Automated Evaluator of WebView

As shown in Figure 3, we develop an evaluator that automatically clicks on links, captures screenshots, and classifies each access to test phishing sites, whether showing the warning page or not.

**Challenges.** Developing an automated testing framework for mobile devices is challenging, as previous research [39] mentioned that finding a suitable emulator was elusive. Moreover, tests within WebView introduce an additional challenge, as each app typically allows only one user account per actual phone number.

Firstly, we overcome these challenges by leveraging the developer options on mobile devices and utilizing the Appium server [2] and the UiAutomator2 driver [3] on the monitoring desktop. When configured, testing URLs are uploaded on apps, and when they are clicked, the evaluator captures the response and automatically saves the screenshots in a folder. Secondly, we address the issue of single-user limitations by issuing additional phone numbers to experimental participants, enabling us to register for each app. Then, we automate various actions signed in by experimental participants on the app, including swiping, clicking, and capturing screenshots.

*3.3.1 Test Suites.* To conduct practical tests, rather than opening links directly on the apps, we have established a simple webpage containing a comprehensive list of test phishing sites accessible only for specific IPs and user-agents. Each mobile device initially accesses the webpage through a dedicated test application, enabling easy access to the phishing site links listed on the GitHub page. Through testing, we observe that the response of each application's WebView varies significantly. For instance, if a phishing site listed on our webpage encounters a blocking action, it displays a warning page and, after activating the back action to access the following phishing site from a warning page, specific apps (e.g., KakaoTalk) return to the messenger interface. In contrast, others revert to the GitHub page. By meticulously examining the distinct behaviors of these applications in preliminary testing, we craft test suite scripts that facilitate the repetitive clicking of phishing site links within each app. They provide precise instructions for link access, navigation, scrolling, and screenshot capture, tailored to the characteristics of each app's WebView.

*3.3.2 Warning Page Classifier.* Figure 2 shows that GSB warning pages, as WSB warning pages for Naver app, displayed in the app's WebView are largely similar to warning pages shown in a Chrome Mobile Browser, except for frames. Consequently, following the methodology of prior research [39], we determine that checking if the dominant color in each image is red is a sufficient criterion for verifying the presence of a GSB warning page. For app-specific warning pages, similar to the methodology mentioned, we record a sequence of test phishing sites' 5 dominant colors (RGB 255) before evaluation and check if the sequence changes because of the apps' warning pages.

*3.3.3 Performance.* We measure the average time it takes for our automation framework to obtain a screenshot of a single test phishing site. In a setup mirroring the conditions of our preliminary test, the average completion time of one access across eight WebViews is recorded as 9.12 seconds. However, when testing with 50 real-world phishing sites, returning to the message window, especially scrolling, consumes lots of time. As a result, across the eight apps, the average time taken is 15.25 seconds, with a maximum of 24.12 seconds. Recognizing this as a challenge in scaleable research, we opt to address it by utilizing multiple devices simultaneously. When we have ran tests on Galaxy Note 7 FE and LG VELVET2, altering only the device information provided in the test suite, we consistently observed an average completion timewith a maximal variance of 1.05 seconds. These results signify that scaleable measurements are attainable with sufficient mobile devices.

## 3.4 Full-Scale Tests

Based on the preliminary test results, we assess the differences between UA-specific attacks targeting WebView and typical phishing attacks on anti-phishing agents, including apps and Safe Browsing. To this aim, we determine the following test scenarios and set the environment of tests.

According to the preliminary test in Section 3.2, GSB demonstrates consistent responses across iOS and Android environments, while reporting through apps depends on the apps themselves and not the mobile OS. Therefore, we conduct the full-scale tests only

| App Name | UA-Specific Phishing | | | Typical Phishing | | | # of Tests |
|---|---|---|---|---|---|---|---|
| | App | GSB | WSB | App | GSB | WSB | |
| KakaoTalk | 10 | 10 | - | 10 | | - | 40 |
| LINE | 10 | 10 | - | 10 | | - | 40 |
| Messenger | 10 | 10 | - | 10 | 10 | - | 40 |
| Facebook | 10 | 10 | - | 10 | | - | 40 |
| Instagram | 10 | 10 | - | 10 | | - | 40 |
| Zalo | - | 10 | - | - | | - | 20 |
| WeChat | 10 | - | - | 10 | - | - | 20 |
| Naver | - | - | 10 | - | - | 10 | 20 |
| **Total** | 60 | 60 | 10 | 60 | 10 | 10 | 210 |

**Table 2: Number of Domains by Test Scenarios**

on Android devices where the automated evaluator can be used. We connect one monitoring infrastructure to one mobile Android device (eight devices) and automate evaluators to check anti-phishing responses every 30 minutes. Given the varying evaluator responses upon accessing the apps and considering factors like errors and delays, we determine that testing with a maximum of 40 domains per app is the most suitable approach. As shown in Table 2, We report 190 domains the built-in reporting function of apps and to GSB. For Naver, we report ten domains per scenario exclusively to WSB.

*3.4.1 Test Scenario.* We decide to test the effectiveness of phishing site detection within the apps by dividing our two scenarios(in Figure 1) into four different subscenarios. We determine each subscenario by where to report phishing links.

**Reporting UA-specific Phishing Sites to Apps.** In this subscenario, we conduct an experiment to confirm whether the apps utilize their own WebView user-agents when attempting to access UA-specific phishing websites. To do this, we input or post each phishing URL link into the app's messaging window or within a post and then report each URL directly to the app. We test seven apps, excluding Zalo, which cannot report phishing links, as mentioned in Section 3.2.

**Reporting UA-specific Phishing Sites to Safe Browsing.** We divide this subscenario based on whether we report to GSB or WSB. We aim to test GSB's ability to detect and block UA-specific phishing sites when reported. We submit our reports through the GSB reporting page [11], designed to provide easy access for genuine reporters. We exclude WeChat and Naver from our experiment since they do not utilize GSB within their WebViews. In the case of the Naver app, we conduct the same test using WSB. However, unlike GSB, we submit our reports via email.

**Reporting Typical Phishing Sites to Apps.** We conduct the experiment in a similar manner to the UA-specific phishing sites mentioned earlier, but this time, we report typical phishing sites.

**Reporting Typical Phishing Sites to Safe Browsing.** As mentioned in the sub-scenario of reporting UA-specific phishing sites to safe browsing, we report typical phishing sites to GSB and WSB in different ways, such as the GSB reporting page and email. We test whether apps exhibit similar reactions to GSB detection as those observed when GSB detects phishing sites. To confirm GSB's detection, we compared the responses in the mobile Chrome browser with those in the apps. As with UA-specific phishing sites reporting to GSB, we exclude WeChat and Naver from the experiment

| App Name | Report to App | | Report to GSB/WSB* | |
|---|---|---|---|---|
| | UA-specific | Typical (Avg. Speed) | UA-specific | Typical (Avg. Speed) |
| LINE | 0/10 | 1/10 (5:05) | 0/10 | |
| KakaoTalk | 0/10 | 6/10 (0:50) | 0/10 | |
| Facebook | 0/10 | 3/10 (2:22) | 0/10 | |
| Instagram | 0/10 | 4/10 (2:30) | 0/10 | 10/10 (0:42) |
| FM** | 0/10 | 3/10 (2:25) | 0/10 | |
| Zalo | - | - | 0/10 | |
| WeChat | 0/10 | 5/10 (4:01) | - | - |
| Naver | - | - | 0/10 | 0/10 |
| **Total** | 0/60 | 22/60 (2:29) | 0/60 | 10/20 |

*: Whale Safe Browsing only for Naver, **: Facebook Messanger

**Table 3: Coverage and Speed of Anti-phishing Entities in WebView (The unit of the average speed is minutes.)**

because they do not use GSB in their WebViews. We conduct the same test with WSB.

*3.4.2 Domain Generation & Hosting Servers.* Considering the responses for each app from the preliminary test results, we conduct experiments with a total of 210 domains, as shown in Table 2. We create and deploy the domains in October 2023, observing responses over 10 hours, based on our preliminary test in Section 3.2 and the previous research [42] showing the average phishing attack lasting 9 hours long to be blocklisted.

We aim to implement our test sites just as phishing sites in the real world. We employ various domain names and multiple web servers to achieve this, striving for diversity. We utilize the DOTHOME [6] hosting service to host eleven Apache web servers, as it permits the connection of up to 20 domains per server. In contrast, we purchase domains from a different registrar, GoDaddy [9], a commonly used choice among phishers. Additionally, we focus on using affordable and commonly employed domains in phishing, such as the '.xyz', '.site', and '.world' domains, frequently used in real-world phishing attacks. To link the domains to web servers, we configure DNS records and use HTTPS, aligning them with conditions similar to phishing sites.

**Test Phishing Brand.** We use the Office 365 login page when creating phishing sites. In 2022, Microsoft was the most frequently utilized brand for phishing kits and was consistently targeted in phishing scams [45, 46]. According to Check Point Research [5], by the second quarter of 2023, Microsoft ranks first on the list of brands most frequently impersonated in phishing scams [18].

## 4 EVALUATION

First, we evaluate the blocking speed and coverage of each test scenario, shown in Table 3. Furthermore, we analyze the logs of our test phishing sites to investigate the strategies employed by anti-phishing entities of WebView when accessing and exploring phishing sites. At last, we examine whether the phishing link sent to each app changes, such as disappearing from the chat room, depending on whether it shows a warning page.

**Crawler Access Attempts.** As shown in Table 4, we have observed that when the crawlers attempt to access UA-specific websites, they use 4.23 times as many different IP addresses and 2.26 times as many diverse user agents compared to when accessing typical phishing sites. However, these access attempts vary between each app and Safe Browsing, with some apps not retrying the access even if the initial attempt failed. It implies that in such apps' WebView environment, any phishing site can potentially evade only the initial

| Phishing Site | Crawler(s) Access | | | | | |
|---|---|---|---|---|---|---|
| | #Domains | IP | UA* | Total | Avg. Access | # of Domains Accessed w/ WebView UA |
| UA-Specific | 130 | 563 | 222 | 758 | 5.83 | 13 (FB**, FM***) |
| Typical | 80 | 133 | 98 | 187 | 1.66 | - |

*: User-agent, **: Facebook, ***: Facebook Messenger

**Table 4: Crawler Access Attempts to our Phishing Sites (Each number represents the count of distinct IPs and user-agents.)**

access of anti-phishing agents, thereby preventing site blocking and extending the lifespan of phishing attacks.

### 4.1 App Performance

Over 10 hours of experiment, as shown in Table 3, 60 UA-specific phishing websites reported to eight apps are not detected when reported through apps with reporting functionality.

*4.1.1 Facebook, Messenger, and Instagram.* When we report typical phishing sites to the apps alone, Facebook, Messenger, and Instagram have less than 50% coverage. Compared to all the sites reported to GSB that have been blocked within 30 minutes, this indicates that, despite having GSB integrated into their WebView by default, the results of reporting to these three apps take longer to reach GSB or may not be transmitted at all.

**WebView User-agent of Facebook Crawler.** When we report the test phishing sites to Facebook, Messenger, and Instagram, the crawlers with the same IPs and user-agent access to test sites immediately after report. Whether reporting a typical phishing site or a UA-specific phishing site, the number of crawlers' access to 3 apps that have accessed the sites is five or more. However, the number of distinct user agents attempting access is 5 for accessing typical phishing sites and 9 for accessing UA-specific sites, indicating the different strategies of inaccessible sites with multiple user agents. Among multiple user agents, we find one crawler of Facebook and Facebook Messenger using their WebView user-agent, which is the only way to access our test sites. As shown in Table 4, 13 UA-specific phishing websites are successfully accessed and crawled by user-agent matching crawler, but the two apps do not take any further action beyond attempting to connect, such as displaying warning messages or alerts.

*4.1.2 KakaoTalk and LINE.* KakaoTalk has ranked the top on both coverage and speed. LINE, the other popular messenger used in South Korea, blocks only one page of the typical phishing site. The crawler of each app has accessed our phishing sites immediately. However, neither of them has used their WebView user-agent, resulting in none of the sucessful access to UA-specific sites. Moreover, their bots only make a single attempt to access both the UA-specific and typical phishing sites reported, no matter whether they fail to access UA-specific phishing sites.

*4.1.3 WeChat.* The app has blocked 5 out of 10 typical phishing sites. We suspect its higher coverage than others happens because it solely depends on its built-in report for anti-phishing.However, it fails to block ten of all UA-specific phishing sites. Furthermore, as KakaoTalk and Line mentioned earlier, WeChat has been observed to attempt to access a site only once after reporting, even if the crawling of sites fails. As of 2023, with approximately 4 million

users in the United States, there is a potential vulnerability for many users.

## 4.2 Anti-Phishing Entities Performance

*4.2.1 GSB.* As shown in Table 3, all typical phishing sites reported solely to GSB are blocked in nearly 30 minutes, almost matching their objective [4] to block malicious sites in 30 minutes. Unlike reporting solely on apps, which shows less than three attempts of access, we find 5.16 access attempts of crawlers on average to access UA-specific phishing sites. Moreover, we have observed more than ten user-agent types when we reported UA-specific phishing sites. Unfortunately, none of the user-agent crawlers with Google's IP include WebView user-agent of apps.

*4.2.2 WSB in Naver.* As with GSB, they all have failed to access our UA-specific phishing sites. We observe an average of only 2.86 access attempts on UA-specific sites, which is not significantly different from the average of 2.25 access attempts on typical phishing sites. WSB alters its operating system and browser during every access attempt, but it has been unable to connect due to the absence of a WebView user-agent, the same as GSB. This fact can be particularly detrimental to apps that rely solely on GSB for phishing URL-blocking mechanisms, such as Zalo, which only blocks reported phishing accounts. In such cases, the apps cannot prevent users from being exposed to phishing links until the phishing accounts are finally blocked.

## 4.3 Links Reported in Apps

We observe whether there are any changes in the status of phishing links posted within the apps based on whether they are reported or not. To this aim, we upload phishing links in each app from the account unrelated to the report at the time both before and after reporting. To better observe how previews change, we have added '*Sign in to your Microsoft account*' on the title of the test phishing site.

**Result.** When we have uploaded the links of UA-specific phishing sites to the apps, all apps have not provided the link preview because the crawlers' user-agents have not included each app's WebView. Additionally, when UA-specific phishing sites have been reported to the apps or the Safe Browsing entity and subsequently uploaded, none of the apps take follow-up measures against reposting these phishing links. Our experimental results indicate that the links of UA-specific phishing sites, which remain unblocked despite reporting, can continue to be used, leading to a continuous risk of phishing for different potential victims after the report. In the case of typical phishing sites, we check that the sites' titles appear in the link previews of Facebook, Messenger, Instagram, KakaoTalk, and LINE before reporting. Even after reporting to apps or the Safe Browsing entity, it is possible to post the same links again or send links, and information indicating that the phishing links are blocked can not be found in the link previews of all apps.

## 5 DISCUSSION AND RECOMMENDATION

While anti-phishing entities are striving to combat evasion techniques of phishing sites, not only on desktops but also mobile platforms [39, 40], our UA-specific phishing site tests on eight apps' WebView illustrate the inadequacy of blocklists against evasion techniques targeted at specific applications. For built-in reports from apps, our experiments reveal persistent disparities in speed and coverage, even when dealing with typical phishing sites. Furthermore, some apps' bots fail to access UA-specific phishing sites but do not attempt to reaccess them, making it unreasonable to rely on the app's built-in reporting for blocking phishing links. These findings highlight phishing site threats in environments that anti-phishing entities have yet to consider. They also emphasize the importance of including the apps' WebView user-agent in the user-agent lists used by their bots.

**Visual Previews of Phishing Links.** Among our tested eight apps, we observe that five apps (excluding Naver, Zalo, and WeChat) provide link previews for received phishing URLs. However, they do not display any warnings or indications for phishing sites on link previews, even when flagged by GSB. Our investigation, involving log analysis of our test sites, find that the bots used by seven of these apps (excluding Zalo) can access links links and to gather crucial information such as HTTPS usage, URL redirection, and the bot's accessibility status. However, this collectable data is underutilized in generating informative link previews and the bots seem to collect only metadata based on the Open Graph protocol [36]. Even these links are abused by phishers [21] for showing plausible images, obscuring phishing links.

As a result, we suggest security recommendations to enhance user protection through more informative link previews. GSB API (Lookup and Update) [22] allows for checking whether a URL is blocklisted and either one is used in WebView by default. By applying this functionality to bots used for link previews, each app's bot can prepopulate link previews based on the link status, whether on a blocklist or not. Moreover, the additional tab for detailed link information, accessible by long-pressing a link, can provide users with valuable insights, including information on HTTPS usage and URL checks via third-party security services. This approach empowers users to make informed decisions before clicking on a link, bolstering their protection against phishing threats in WebView.

## 5.1 Ethics

We consider several ethical issues, while conducting the study.

**Responsible Disclosure.** We have disclosed our experimental findings to relevant Safe Browsing entities and apps. Specifically, we have reported that specific phishing sites accessible only through the WebView of a particular app are not adequately mitigated by Google Safe Browsing, and we are awaiting a response. Regarding the apps, we have reported similar contents to GSB for KakaoTalk, Line, Instagram, and Facebook Messenger. We note that the bots of each app cannot reach the accurate phishing site that can only be accessed through each app's WebView. In the case of Facebook, we report that Facebook's bots can access phishing sites through Facebook and Messenger's WebViews. However, we have not encountered any phishing site warning pages during the experiment. Additionally, we have shared our findings with applications that either possess their own safe browsing mechanisms (e.g., Naver) or have no discernible mechanisms in place (e.g., WeChat) based on our experiments.

**Risk of Benign Access.** In order to prevent potential victims from accessing our created test phishing sites, the URLs of these phishing

sites have been exclusively reported to anti-phishing entities and have not been uploaded to other services. Furthermore, access to our website containing the list of phishing sites used for automated phishing site access has been restricted to authorized experimental participants only. Lastly, as a precautionary measure against benign access, we refrain from conducting additional server-side processing for any form submissions and stop hosting test phishing sites immediately after the experiment period.

**Infrastructure Usage.** We followed the guides for all the services used for our study. Also, we have informed our hosting service, Go-Daddy and DOTHOME, of our experiments, and they have allowed us to utilize their infrastructure.

## 5.2 Limitation

**Partial Automation.** In our experimental framework, we manually acquire domains and host web servers. However, with the future implementation of a secure process for automated test phishing site generation, our framework is poised to become a comprehensive automation platform for evaluating app-targeted phishing within WebView. Notably, through the use of simple test suites in our framework that determine behaviors for each application, safe-browsing developers can also assess the anticipated blocking outcomes of apps that employ their own safe browsing mechanisms.

**Short-term Experiment.** We observe the effects of reporting for a limited period of 8-10 hours after the reporting incident. This limitation prevents us from observing the possibility of further blocking by apps or Safe Browsing for our test phishing sites. However, as mentioned in previous research [42], phishing attacks in the real world take an average of 9 hours to be blocklisted after the initial visit, indicating that our measurement period captures practical information on anti-phishing behaviors.

## 6 RELATED WORK

As far as we know, this paper represents the first investigation into the response of the anti-phishing ecosystem of WebView when user-agent evasion techniques are employed for targeting apps' WebView. It is likely the first research paper to develop an automation framework for evaluating anti-phishing detection within the mobile app environment. While similar studies [39, 40, 48] exist, they predominantly focus on research conducted in the mobile browser environment, with most research concentrating solely on phishing site cases within the desktop environment.

**Measuring Anti-Phishing Ecosystem.** Oest et al. [39] introduced PhishFarm, a scalable framework designed to systematically assess the effectiveness of blocklisting coverage and timeliness in modern desktop and mobile browsers when faced with evasion techniques employed by attackers. During a period spanning from mid-2017 to late 2018, Google Safe Browsing (GSB) blocklists exhibited significant issues on mobile devices, both with and without evasion techniques. Notably, none of the websites triggered any warnings in mobile Chrome, Safari, or Firefox, despite being blocklisted on desktop platforms. However, following the disclosure of their findings, there is now greater consistency in blocklisting practices between desktop and mobile platforms compared to the earlier period.

Oest et al. [40] examined the speed of blocklisting on mobile devices. This examination involved the programmatic monitoring of Google Safe Browsing through the Update API. Additionally, the study empirically compared the mobile versions of Chrome, Firefox, and Opera with their desktop counterparts, utilizing a single physical Android phone.

**WebView Security.** Android WebView is vulnerable and susceptible to various attacks. Luo et al. [38] and Chin et al. [33] explored the vulnerabilities within webviews and various attacks. Yang el al. [47] made an empirical study that when an untrusted web iframe/popup is present within a WebView, it can bypass existing defense solutions and potentially lead to the leakage of sensitive information, potentially resulting in phishing attacks. Zhang et al. [50] performed the first empirical study on the usability in security of in-app browsing interfaces (IABIs) in both Android and iOS apps. The information on In-app browser is limited and only a few In-app browsing interfaces give warnings to remind users of the risk of inputting passwords during navigating a login page which can lead to phishing attack.

However, we further discover that earlier studies exclusively examine the performance of phishing entities in major mobile browser environments, and these improvements have not been effectively applied in WebView browsers. Given the growing trend of mobile apps incorporating WebView-based in-app browsers, it is imperative to assess their effectiveness in applications utilizing WebView browsers.

## 7 CONCLUSION

We have conducted a first systematic evaluation of anti-phishing entities blocking processes in individual apps' WebView in real-world scenarios, specifically tailoring phishing attacks to WebView. We have targeted eight well-known apps utilizing WebView, reporting 50 general phishing sites and 80 user-agent-specific (UA-specific) phishing sites. The results of our comprehensive analysis reveal that UA-specific phishing sites evade blocking measures across all eight Android apps. By logging access to test phishing sites, the crawlers attempt to access UA-specific websites with 4.23 times more IP addresses and 2.26 times more diverse user agents than when accessing typical phishing sites, implying anti-phishing agents' strategies to reach the real phishing site. In spite of anti-phishing crawlers' efforts to access, we find that only two apps identify their accessible user-agents for the phishing site, not taking further actions such as showing warning pages. Moreover, we observe that UA-specific phishing links in apps do not provide any information on link previews, even after they are reported, being able to be the constant threat of phishing. The findings of this study underscore the need for developing and implementing more evident visual cues for users when accessing websites through WebView. It is imperative that future work builds upon these findings to create a safer WebView experience for users and provide more effective approaches for preventing phishing threats.

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
