# OpenReview forum: "PhishinWebView: Analysis of Anti-Phishing Entities in Mobile Apps with WebView Targeted Phishing"
_ACM.org/TheWebConf/2024/Conference — TheWebConf24_

### Official Review · Reviewer_SBXq · 2023-11-10

**Novelty:** 5
**Technical Quality:** 6

**Review:**

This paper proposes a phishing cloaking attack against mobile app Webviews. The cloaking idea is that the phishing site only reveals phishing content when the user-agent matches that of the target app’s Webview. The paper evaluates how Google Safe Browsing (GSB) and in-app anti-phishing systems react to such phishing sites after the phishing URLs are reported. The results show that current anti-phishing engines cannot handle such cloaking well.

Strength

-	The first study to investigate Webview user-agent-based cloaking for phishing
-	It is interesting to know popular messaging apps have built-in anti-phishing solutions
-	Responsible disclosure of the discovered problem

Weaknesses

- Small-scale study (8 apps). The core of the problem is the diversity of Webview user-agent and the study does not have corresponding results
- The automated test suite cannot support large-scale tests
- The security recommendation is unclear; needs clarification.


Overall, the proposed attack is interesting. Like other cloaking techniques, the key is for the phishing web server to selectively react to different visitors. In this case, the phishing server only returns malicious content to specific Webview user agents. The findings are interesting, that is, phishing URLs are not blocked even after user reporting. However, the fundamental issue is still cloaking. After user reporting, security vendors (e.g., GSB) cannot use the correct user-agent to visit the reported URLs and thus cannot verify the phishing content.

**Solution and Recommendation**
I am not sure if the recommendation from the paper makes sense (page 7). The authors argue it should enhance the link previews and give users information about HTTPS usage and the results of URL checks from third-party services. This is confusing on many levels. First, HTTPs usage is a known misconception in phishing detection. Phishing sites these days often use HTTPs to get the correct certificate (the green padlock) to present a false sense of security to users. It is a wrong signal for phishing detection. More importantly, the problem is third-party services do not have the correct user agent to visit this URL. The solution seems to be either (1) specific apps use their own user agent to visit the URL and perform phishing checks (like what Facebook did), or (2) third-party security services are provided with the app-user agent to crawl the URL.

**App-specific User Agent, Study Scale**

The fundamental problem is whether there is a large variety of app-specific user agents that GSB should include in their phishing crawler. Again, the phishing content needs to be checked before the URL can be admitted to the blocklist. I was hoping to see some analysis of this core issue but unfortunately, there is no such analysis. For example, is it possible to download Top-K popular apps from the market (K=some large number, e.g., 10,000), and automatically extract the Webview user agents, and provide them to GSB (or other security vendors) to enhance their crawler? The current analysis is limited to 8 apps focusing on messaging, but the problem is beyond messaging apps --- any apps with a Webview and allow for user-generated content are subject to this attack.

**Test Suite**

The authors investigate significant efforts to automate the test of 8 apps. However, it is difficult for the test suite to automatically scale to test many apps because each app’s script needs to be manually customized. Do you see a way to scale up the tests with such automation (to cover more apps)? For instance, if we only need binary answers (blocked or not) instead of how quickly a URL get blocked, would it be possible?

**Clarification**

Section 4.1: "One crawler of Facebook and Facebook Messenger using their WebView user-agent" – this means Facebook was able to get the phishing site correctly, but the two apps did not take any further action. What is the reason?

**Questions:**

- Can you clarify the recommendations on page 7?

- Is there a large variety of app-specific user agents that GSB should include in their phishing crawler? Can we extract such user-agents from a large number of apps on the market?

- Section 4.1: "One crawler of Facebook and Facebook Messenger using their WebView user-agent" – this means Facebook was able to get the phishing site correctly, but the two apps did not take any further action. What is the reason?

**Ethics Review Description:**

The authors discussed their risk mitigation methods.

**Reviewer Confidence:**

4: The reviewer is certain that the evaluation is correct and very familiar with the relevant literature

**Scope:**

4: The work is relevant to the Web and to the track, and is of broad interest to the community

---

### Official Review · Reviewer_1omU · 2023-11-23

**Novelty:** 6
**Technical Quality:** 5

**Review:**

This paper provides a systematic evaluation of anti-phishing entities within real-world in-app browsers (i.e., WebView environments). The study encompasses Google/Whale safe browsing entities and built-in anti-phishing functionality. To achieve comprehensive results, the authors develop an automated evaluation framework for Android WebView and test eight popular social and messenger apps using 210 phishing websites.

**Pros**
+ The paper is the first to comprehensively study and evaluate anti-phishing entities in WebView environments.
+ An automated framework is designed to evaluate anti-phishing properties in eight apps.
+ The authors introduce a new UA-specific attack on WebView environments and measure the reactions of anti-phishing entities in eight apps to this attack.

**Cons**
- The limitation on warning page classifiers needs clarification.
- The study scale may be limited.
- Some design details of the automated evaluator are missing (refer to Question 2-3).
- The presentation in certain parts lacks organization and clarity (details listed in the Questions section).

**Minor**
- There are grammatical mistakes and typos, I list a few in the following:
1) On page 5, “As shown in Table 2, We report 190 domains the built-in reporting function of apps” should be “As shown in Table 2, we report 190 domains to the built-in reporting function of apps”.
2) On page 7, “find that the bots used by seven of these apps (excluding Zalo) can access links links” should be “find that the bots used by seven of these apps (excluding Zalo) can access links”.

**Questions:**

1) Why is it reasonable for warning page classifiers to use 5 dominant colors as classification indicators?
The paper should justify this choice in Limitation, considering the possibility of similarity between the dominant colors of normal and warning pages.

2) Why should we use multiple user accounts in the automated evaluator?
Section 3.3 lacks clarification on why multiple phone numbers are needed to register multiple user accounts.

3) How do the authors utilize the automated evaluator to “examine whether the phishing link sent to each app changes”?
The paper mentions examining changes in sending phishing links before and after reporting, but the design details of this process are not provided. Is it a manual or automated process?

4) Organizational and clarity issues.
  a. The term 'crawler' in the Introduction needs an earlier explanation for improved readability.
  b. Clarification is needed regarding the statement, "As shown in Table 2, We report 190 domains to the built-in reporting function of apps and to GSB."  How "190 domains" are computed?
  c. In Section 4, the term 'failed' is ambiguous. It should be clarified whether it refers to the apps being unable to access websites or access phishing websites.

**Ethics Review Description:**

NULL

**Reviewer Confidence:**

3: The reviewer is confident but not certain that the evaluation is correct

**Scope:**

4: The work is relevant to the Web and to the track, and is of broad interest to the community

---

### Official Review · Reviewer_URjE · 2023-11-23

**Novelty:** 4
**Technical Quality:** 4

**Review:**

**Strengths**:
* New study that attempts to evaluate the ability of anti-phishing entities to detect WebView-targeted phishing.
* Well-written paper that addresses an important topic with real-world significance.

**Weaknesses**:
* The study only focused on UA based evasion: there are many other attack vectors that an attacker might use to fingerprint and target web-view specific devices.
* Reporting setup might not reflect real-world scenarios: only one report from one source for each candidate phishing domain with no attempts at diversified reporting.
* Lack of in-the-wild phishing results to complement this study and improve its strength.

**Detailed comments**:
I thank the authors for working on this submission. I enjoyed reading the paper as it attempts to address an important issue. I want to add a note that I have personally experienced a UA-only based phishing attack on Facebook. The phishing attack in question was targeting Facebook’s login page itself. I confirmed the evasiveness of this phishing site by visiting it from outside of Facebook’s Webview browser which no longer led me to the phishing payload. It was interesting to note that this phishing payload was delivered to me by a friend of mine whose account was phished by the same technique thus showing the interesting “viral nature” as well as the real-world nature of webview-based phishing attacks. I am sharing this story here with the authors to reaffirm the direction of research chosen to be pursued by the authors and commend them for it.

At the same time, I have to say that the technical depth presented in this study is a bit too shallow currently especially in light of prior works. Below, I will enumerate the weaknesses I perceived in this study and also a discussion for each about how these can be improved to potentially result in a much stronger submission in the future.

1. **Lack of variety in evasion tactics**: This study tried to see how successfully an attacker who is focused on evading Anti-Phishing-Entities on WebView-based URLs can pursue such evasion. However, the evasion tactics were limited exclusively to UA-based evasions i.e. the attacker would serve phishing content to any client that presents User-Agent headers that match the targeted Web-View client but benign content (or, was it 404 error?) otherwise. However, there are so many other tactics that the attacker can pursue for figuring out if a client is coming from a WebView-based browser. For example, there might be other headers that might be giving this information away. The presence of certain mobile-specific APIs (such as Gyroscope, Accelerometer, Touch APIs etc.) can all be used as evidence for the presence of a mobile-specific client as opposed to simply relying on UA field. These are arguably much harder for an anti-phishing entity to fix as opposed to the simple UA field. I suggest the authors to read [A] which focused on this topic for more insights for broadening this study. Furthermore, another work, [B] also tried to go beyond UA-headers by focusing on computing browser fingerprints of anti-phishing entities as well as visitors in an attempt to evade vetting engines. While none of these works have in deed reported their URLs to In-app browsers as was done in this study, these works give an idea about how to go beyond just UA-based evasion and systematically generalize the pool of evasion vectors potentially available for the attackers to utilize. Such “systematization” of potential evasion tactics can immensely help the authors of this study as well. One simple method to do this is to set up a small “profiling” experiment in which the authors themselves build a few benign honeypot websites as was done in these prior works and self-report them to analyze all headers and fingerprints that can be collected from in-app URL vetting engines deployed by the apps. This results of this experiment can then “inspire” the authors to add more depth to their phishing experiments.
2. **Insufficiency of reporting**: The authors appear to have reported each URL only one time from only one source. Unfortunately, this does not suffice for real-world scenarios. For example, in order to deal with potential false reports,  the in-app URL vetting engines might likely only act upon reports that they receive X amount of times from different users. Prior works such as PhishFarm as well as [A] have pursued tactics such as “repeated reporting” from “diverse sources” in order to account for this. I suggest the authors to pursue similar tactics as well in order to improve the readers’ confidence in their results.
3. **Lack of in-the-wild datasets**: In another direction, the authors should also consider collecting real-world datasets of in-the-world phishing attacks targeting web-view clients as was mentioned at the beginning of this review. Use of datasets such as those related to real-time “Certificate-Transparency” data can serve as a potential rich source for attempting to collect this kind of data. Such real-world measurements can set this paper apart from several prior works such as PhishFarm, [A] and [B] all of which focused only on artificial author-generated sites.
 On a minor note, it appeared that the evasion tactics also need some improvements. For example, how exactly were the UA-specific websites setup when the visitors come from non-app clients? Would they be served a 404 page or would they be served a benign web page. From an evasion perspective, may be serving a benign page would be better as serving a 404 page can potentially solicit more visits from security entities which is unnecessary from the perspective of the attackers.

[A] A Human in Every APE: Delineating and Evaluating the Human Analysis Systems of Anti-Phishing Entities, DIMVA 2022
[B] PhishPrint: evading phishing detection crawlers by prior profiling, USENIX Security 2021

**Post-discussion update**: As a result of very positive discussion with the authors, I am increasing my scores for this paper and would vote for its acceptance if the other reviewers agree.

**Questions:**

1. Were each of the URLs reported only once?
2. Were the visitors from non-targeted apps served 404 pages or benign content (with 200 status code)?

**Reviewer Confidence:**

4: The reviewer is certain that the evaluation is correct and very familiar with the relevant literature

**Scope:**

4: The work is relevant to the Web and to the track, and is of broad interest to the community

---

### Official Review · Reviewer_ZSqP · 2023-11-23

**Novelty:** 4
**Technical Quality:** 3

**Review:**

**Summary**
The paper primarily focuses on the anti-phishing process in webview in a set of popular individual apps. The paper introduces an analysis pipeline to investigate how users might get exposed to phishing websites that target useragents of webviews.
The analysis shows that only a few apps implemented the safebrowsing effectively.


**Strengths**

1 - An analysis of phishing attacks on webview is interesting.
2 - The security implications can impact many users, assuming the apps being used in this study are among the most popular messaging  mobile apps.



**Weaknesses**
1 - The core security insights are a bit abstract. The descriptions of the problem and findings can be improved
2 - The takeaways and main findings need to be highlighted.

**Detailed Comments**

I would like to thank the authors for defining a project in this area. Webview is an important part of apps these days and security analysis of the service, from different perspectives, are indeed important. That being said, I believe the paper can be improved in a few different ways. I understand that the paper used those 8 applications based on their popularity in specific regions, but it is not clear to me how the issue can be generalized. Is the issue also relevant to other forms of apps that also allow messaging (e.g., Zoom, slack, etc.). It seems the issue is not specific to messaging app and every app that uses webview might have the issue. If this is the case, it needs to be highlighted in the discussion, app selection criteria.
Also, it is not clear what would be the role of app developers in this process. I understand that offering some better visualization could help users, but what needs to be done in the background that is not happening effectively now.

I appreciate the authors' effort in defining an ethics procedure for their study. However, it was not clear what has been acknowledged. I understand this part has never been easy, but we are talking about important apps that are being used by a large population. Providing greater details on the dynamics of finding exchanges and the results seems critical (How early the results were shared before submission, what were the responses, how the findings were perceived?)

Lastly, tables and figures need more explanations. It was a bit difficult to understand the content of the table and their takeaway. For instance in table 2 it is not clear how the reader should interpret the findings. How GSB is different in UA-Speific phishing vs typical phishing? Both report the number 10, but in two different format. Also the text says, “
As shown in Table 2, We report 190 domains the built-in reporting function of apps and to GSB”. I was not sure where those 190 domains were shown in the table.


I think this is an interesting paper with an interesting direction, but the findings need to be highlighted more objectively. the analysis pipeline and the app selection criteria need to be discussed more effectively. The discussion needs to reflect what is it that apps’ developers with webview capabilities need to do that they are not doing now.

**Questions:**

Please read the reviews on ethics, app selection criteria, and discussions on genelizability of the observations.

**Ethics Review Description:**

It seems that the authors have contacted the apps, but the details of the procedure is not clear.

**Reviewer Confidence:**

4: The reviewer is certain that the evaluation is correct and very familiar with the relevant literature

**Scope:**

4: The work is relevant to the Web and to the track, and is of broad interest to the community

---

### Official Review · Reviewer_Mtge · 2023-11-26

**Novelty:** 4
**Technical Quality:** 5

**Review:**

This paper studies anti-phishing protection of mobile apps that use WebView to view URLs. The authors crafted test phishing websites and implemented an evaluation framework to measure accessibility of these websites via app WebViews. In the results, they found that the User-Agent specific phishing sites (i.e. websites that show phishing content only in case of a WebView related User-Agent) successfully evade blocking across all the eight apps they analyzed. Moreover, anti-phishing crawlers (from app developers and Google Safe Browsing) are prevalently not using specific User-Agents that are necessary to detect these phishing websites.

Pros. The paper provides interesting technical details on the WebView cloaking attacks, as well as highlights the limitation in anti-phishing solutions against cloaking. In fact, I was surprised by the Facebook Crawler examples in section 4.1. The study is clear and the paper is well written. The authors provide security recommendations and there are definitely straightforward action items for app and WebView developers, as well as for anti-phishing solution providers to improve protection of their users. And the authors already stated that they reported their findings to the anti-phishing entities.

Cons. The main weakness of the paper is the lack of generalization of the attack. Cloaking is a known big issue (e.g., prior work already showed this issue for mobile browsers), and I can imagine more similar studies, when researchers instead of WebView UA, can simulate attacks targeting other specific UAs - and it is very likely such attacks can bypass anti-phishing crawlers as those cannot crawl with all possible UAs. As such, I would advise the author to discuss other applications of the similar attack structure in the paper. I would also expect a comparison with desktop UA cloaking experiments.

On the other hand, given the generic cloaking issue, anti-phishing entities do not only operate only the crawled data and take into account other signals of a domain or a URL, such as lexical patterns, popularity in the user traffic, reputation of IP, variety of recipients, credential exfiltration endpoints, etc. This is a potential technical limitation of the current study, as it does not test a real phishing campaign, but simulates phishing pages instead. I would like to find out more details about the test domains, and I would advise the authors to analyze real phishing campaigns in addition to specifically crafted test pages. If you have seen similar attacks in the wild, this would help with motivating the paper.

Smaller issues:
- The scale of the experiment doesn't look big (only 8 apps, only Android platform, only one phishing content for Office 365, etc.). It would be nice to have a discussion on the technical feasibility to use the same automation framework for more experiments.
- Why only wait for 8-10 hours and not evaluate for a longer period of time? It would be very interesting to find out at what time the coverage is being added.
- In addition to Safe Browsing, it would be interesting to test other URL filtering and categorization providers.
- Section 6 mentions that “there is now greater consistency in blocklisting practices between desktop and mobile platforms”, which requires a reference or more explanation.

**Questions:**

Could you provide more details on the phishing pages used for evaluation? Namely, whether domains had any phishing-like patterns, any certificates used, and what was the actual behavior in case of the non-WebView UA.

**Reviewer Confidence:**

4: The reviewer is certain that the evaluation is correct and very familiar with the relevant literature

**Scope:**

4: The work is relevant to the Web and to the track, and is of broad interest to the community

---

### Decision · Program_Chairs · 2024-01-22

**Decision:**

Accept

**Comment:**

# Summary

 This paper studies an interesting new type of phishing attack, wherein an attacker shares a link to a user on a mobile app, and the mobile app opens the link using the built-in WebView. Then, the phishing website can easily evade detection by detecting the User-Agent (which is unique to the mobile app). The paper measures the effectiveness of the anti-phishing ecosystem in detecting these types of evasions.

 # Strengths

 + Interesting new phishing attack.
 + Automated system to test mobile app UA-based phishing evasion.
 + Paper provides security recommendations, and authors reporting findings to the app developers.

 # Weaknesses

 - Limited focus: just on WebView UA cloaking effectiveness for phishing.
 - Unclear real-world applicability: not yet found an in-the-wild instance (although one reviewer reported an anecdotal instance).

 # Recommendation

 Overall, the reviewers appreciated that this paper addresses a novel and interesting attack vector in phishing. Many were surprised---and one reviewer had first-hand experience with---by WebView-driven phishing and the related cloaking. The authors described additional experiments that the authors in the discussion phase, and these results will greatly strengthen the paper.

 ---